# Organic electrochemical transistor arrays for real-time mapping of evoked neurotransmitter release in vivo

Kai Xie[1†], Naixiang Wang[2†], Xudong Lin[1], Zixun Wang[1], Xi Zhao[1], Peilin Fang[1], Haibing Yue[1], Junhwi Kim[1], Jing Luo[3], Shaoyang Cui[3], Feng Yan[2*], Peng Shi[1,4,5*]

[1]Department of Biomedical Engineering, City University of Hong Kong, Kowloon, China; [2]Department of Applied Physics, Hong Kong Polytechnic University, Kowloon, China; [3]Department of Rehabilitation, Shenzhen Hospital of Guangzhou University of Chinese Medicine, Shenzhen, China; [4]Shenzhen Research Institute, City University of Hong Kong, Shenzhen, China; [5]Center of Super-Diamond and Advanced Films (COSDAF), City University of Hong Kong, Kowloon, China

**Abstract** Though neurotransmitters are essential elements in neuronal signal transduction, techniques for *in vivo* analysis are still limited. Here, we describe an organic electrochemical transistor array (OECT-array) technique for monitoring catecholamine neurotransmitters (CA-NTs) in rat brains. The OECT-array is an *active* sensor with intrinsic amplification capability, allowing real-time and direct readout of transient CA-NT release with a sensitivity of nanomolar range and a temporal resolution of several milliseconds. The device has a working voltage lower than half of that typically used in a prevalent cyclic voltammetry measurement, and operates continuously *in vivo* for hours without significant signal drift, which is inaccessible for existing methods. With the OECT-array, we demonstrate simultaneous mapping of evoked dopamine release at multiple striatal brain regions in different physiological scenarios, and reveal a complex cross-talk between the mesolimbic and the nigrostriatal pathways, which is heterogeneously affected by the reciprocal innervation between ventral tegmental area and substantia nigra pars compacta.

*For correspondence:
apafyan@polyu.edu.hk (FY);
pengshi@cityu.edu.hk (PS)

†These authors contributed equally to this work

**Competing interests:** The authors declare that no competing interests exist.

## Introduction

In the nervous system, neurotransmitters (NTs) are released upon arrival of action potentials, and play essential roles in signal transmission for regulating physiological activities (*Kavalali, 2015*). Though electrical recording and stimulation of neural activities have offered extremely powerful and widely used toolsets for neuroscience research, electrophysiological methods typically lack cell selectivity and can only be used for stimulatory but not inhibitory manipulation, and their spatial resolution can be limited and highly depend on the size of electrodes (*Tye and Deisseroth, 2012*). Alternatively, the methods for the direct measurement of NT release are relatively limited (*Wightman and Robinson, 2002*), partially due to the complex chemical dynamics and a substantial variety of NTs. Among the large family of NTs, the category of catecholamine neurotransmitters (CA-NTs) consists of dopamine, noradrenaline, and adrenaline, which all share a common chemical structure containing a catechol with two hydroxyl groups and a side-chain amine. These NTs influence a great variety of neural functions (*Fields et al., 2007*), and have been one of the major targets for previous efforts on the detection of NTs in animal brains (*Wightman and Robinson, 2002*; *Bucher and Wightman, 2015*).

Currently, a few techniques are available for the detection of NT release *in vivo*. Microdialysis-based method is a two-step analytical process that involves sample collection from the brain and a later analysis by liquid chromatography (*Fiorino et al., 1993*). It can only resolve averaged

**eLife digest** Cells in the nervous system pass messages using a combination of electrical and chemical signals. When an electrical impulse reaches the end of one cell, it triggers the release of chemicals called neurotransmitters, which pass the message along. Neurotransmitters can be either activating or inhibitory, determining whether the next cell fires its own electrical signal or remains silent. Currently, researchers lack effective methods for measuring neurotransmitters directly. Instead, methods mainly focus on electrical recordings, which can only tell when cells are active.

One new approach is to use miniature devices called organic electrochemical transistors. Transistors are common circuit board components that can switch or amplify electrical signals. Organic electrochemical transistors combine these standard components with a semi-conductive material and a flexible membrane. When they interact with certain biological molecules, they release electrons, inducing a voltage. This allows organic electrochemical transistors to detect and measure neurotransmitter release. So far, the technology has been shown to work in tissue isolated from a brain, but no-one has used it to detect neurotransmitters inside a living brain.

Xie, Wang et al. now present a new device that can detect the release of the neurotransmitter, dopamine, in real-time in living rats. The device is a miniature microarray of transistors fixed to a blade-shaped film. Xie, Wang et al. implanted this device into the brain of an anaesthetised rat and then stimulated nearby brain cells using an electrode. The device was able to detect the release of the neurotransmitter dopamine, despite there being a range of chemicals released inside the brain. It was sensitive to tiny amounts of the neurotransmitter and could distinguish bursts that were only milliseconds apart. Finally, Xie, Wang et al. also implanted the array across two connected brain areas to show that it was possible to watch different brain regions at the same time.

This is the first time that transistor arrays have measured neurotransmitter release in a living brain. The new device works at low voltage, so can track brain cell activity for hours, opening the way for brand new neuroscience experiments. In the future, adaptations could extend the technology even further. More sensors could give higher resolution results, different materials could detect different neurotransmitters, and larger arrays could map larger brain areas.

biochemical events over minutes and has relatively poor spatial resolution. Electrochemical techniques are developed to resolve transient biochemical processes. Amperometry has the advantage of generally good detection limits but is restricted to electroactive species, and is not suitable for detection in complex environment (*Bucher and Wightman, 2015*). Cyclic voltammetry (CV) was adopted to monitor biochemical fluctuation in mammalian brains by recording oxidation-induced changes in related current-voltage curves (*Kissinger et al., 1973*; *Robinson et al., 2003*). While this technique is efficient and NT selective, the readout from CV is not intuitive, and the technique also involves a complex instrument configuration (*Atcherley et al., 2015*). So far, it is still challenging to have a fully implantable multi-channel device that can detect and monitor NT release with sufficient sensitivity and temporal-spatial resolution. Recently, genetically encoded fluorescent sensor has been developed for detecting dopamine *in vivo* (*Patriarchi et al., 2018*), which has improved selectivity and temporospatial resolution. But it also requires genetic modifications along with complex optical recording equipments, and is less quantitative compared to other methods.

Organic electrochemical transistor (OECT) has emerged as a promising transducer for detecting chemical, electrical, and molecular signals (*Rivnay et al., 2018*; *Rivnay et al., 2017*; *Lin and Yan, 2012*; *Lin et al., 2011*; *Tang et al., 2011b*; *Khodagholy et al., 2013*), in virtue of its intrinsic amplification capability, low operating voltage, mechanical flexibility, and efficient ion transport/exchange between the device and electrolyte environment (*Lin and Yan, 2012*). So far, many biological molecules (including NTs) and cellular activities have been successfully detected *ex vivo* by using the OECT platforms (*Kergoat et al., 2014*; *Rivnay et al., 2015*; *Jonsson et al., 2016*; *Campana et al., 2014*). However, considering the complex biological environment, recent efforts of *in vivo* biosensing by OECTs mostly focus on stimulation or detection of electrophysiological signals (*Williamson et al., 2015*; *Lee et al., 2016*), while the release and transport of chemical substances in living organisms receive less investigations (*Li et al., 2015*; *Xu et al., 2016*). In this study, we describe a fully implantable OECT-array for *in vivo* detection of NTs. This device is capable of

multichannel monitoring of dopamine release as low as 1 nM with a 50 ms sampling rate, and is well suitable for usage in living animals. Using this OECT-array, we successfully demonstrated real-time, multi-site, simultaneous monitoring of CA-NT release at different brain regions, including ventral tegmental area (VTA), nucleus accumbens (NAc), and caudate putamen (CPu), in response to neural stimulation along dopaminergic pathways. Our study provides an electrochemical mapping in the striatal brain and also reveals a complex cross-talk between mesolimbic and nigrostriatal signalling, which relies on a reciprocal innervation between VTA and substantia nigra pars compacta (SNc) nuclei (*Margolis et al., 2008*). We found that the connection between the two nuclei can significantly affects dopamine release in NAc and lower CPu, but not upper CPu, suggesting a heterogeneous mapping from SNc to CPu (*Lerner et al., 2015*).

## Results

### Working principle of OECT-array for NT detection

Our device is based on a miniaturized microarray of OECTs (OECT-array) on a 200 μm thick polyethylene terephthalate (PET) substrate. Each functional unit consists of a platinum GATE (Pt-GATE) electrode and a conductive poly(3,4-ethylenedi-oxythiophene):poly(styrene-sulfonate) (PEDOT:PSS) film that bridges the gold SOURCE and DRAIN electrodes (*Figure 1a*). When an OECT-array was engaged in a biological electrolyte, some molecules (e.g., CA-NTs; *Figure 1b*) could undergo electro-oxidation reaction on the surface of the Pt-GATE electrode. In this process, the catechol groups of CA-NT molecules are oxidized into quinone with the release of two electrons, which are

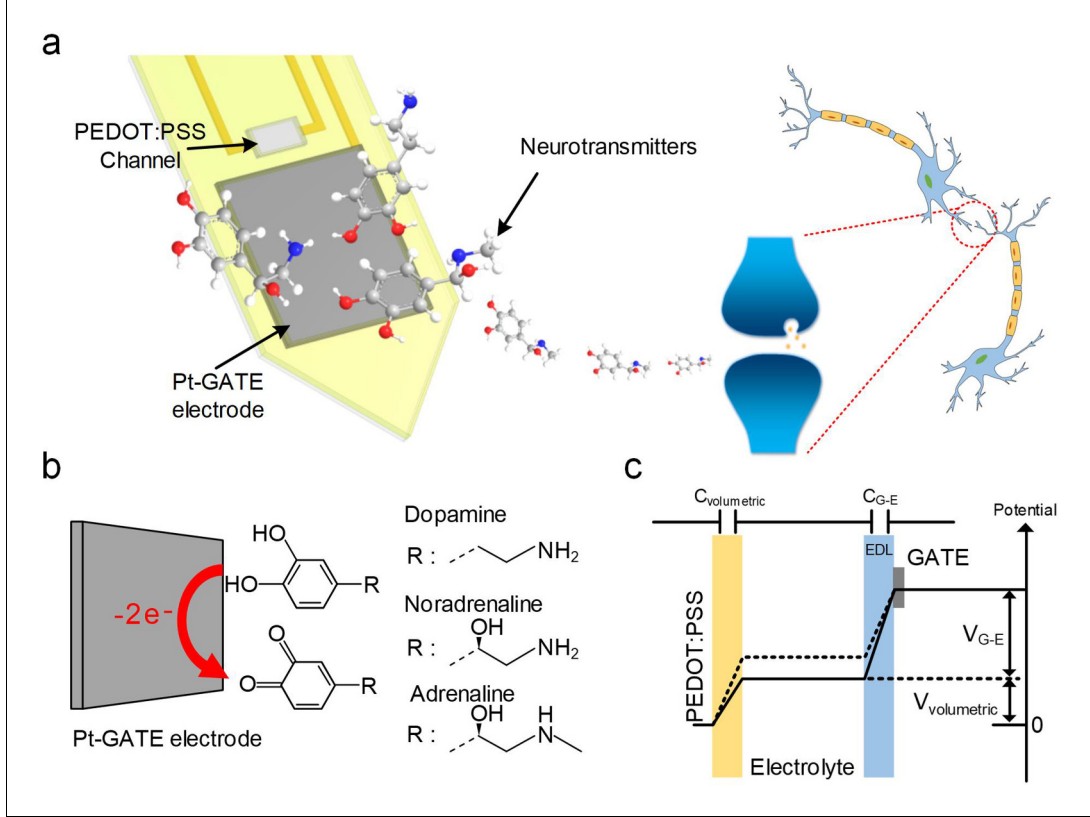

**Figure 1.** System schematic diagram and OECT-array working principle for CA-NT detection. (**a**) Systematic diagram of using the OECT for monitoring neurotransmitter release. (**b**) Illustration of CA-NT's electro-oxidation reaction on the surface of the Pt-GATE electrode. (**c**) Diagram showing the working principle of an OECT device. $C_{volumetric}$ and $C_{G-E}$ denotes the volumetric capacitance across the PEDOT:PSS active layer and the capacitance between the GATE electrode and the electrolyte respectively. $V_{volumetric}$ and $V_{G-E}$ denotes voltage across the active PEDOT:PSS layer and between the GATE electrode and the electrolyte respectively.

transferred to the GATE to generate a Faradic current, decrease the potential drop at the GATE/electrolyte interface, and subsequently increase the effective GATE voltage $V_{g-eff}$, as described by:

$$\Delta V_{g-eff} \propto 2.30(1+\gamma)\frac{\kappa T}{2q}log[C], \tag{1}$$

where $\gamma$ is the ratio between the volumetric capacitance of the PEDOT:PSS active channel ($C_{volumetric}$) and the capacitance of the GATE-electrolyte interface ($C_{G-E}$) (*Figure 1c*); $k$ is the Boltzmann constant; $T$ is the absolute temperature; q is the charge of an electron; [C] is the concentration of the reactive molecules released into the biological solution (*Tang et al., 2011a*). Notably, *Equation (1)* is normally applicable in a relatively high-concentration regime; For lower concentrations of analyst, we can use an empirical relationship given by *Zhang et al. (2014)*:

$$\Delta V_{g-eff} = A \times [C]^{\beta}, \tag{2}$$

where $A$ and $\beta$ are constants determined by fitting our experimental data.

## Electrical characterization of the OECT device

To accommodate an easy implantation in the brain of a living animal, the OECT-array was designed and fabricated into a slim blade shape (~1 mm wide, ~15 mm long, and ~200 μm thick) with a tapering tip. Four sets of functional OECT units were integrated at the top 5 mm of a device, each spaced by 1.2 mm (*Figure 2a*, also see *Figure 2—figure supplement 1*). The Pt-GATE and the PEDOT:PSS channel were exposed to the biological environment and all the rest area was insulated by a micro-patterned layer of SU-8 photoresist. In the electrical characterization, a dual channel sourcemeter was used to provide the GATE voltage between the Pt-GATE and the SOURCE electrode ($V_{GS}$), and the drain voltage across the active channel ($V_{DS}$, the voltage between the DRAIN and the SOURCE electrode); the channel currents running through the DRAIN and the SOURCE electrode ($I_{DS}$) was monitored (*Figure 2b*). A transfer curve ($V_{GS}$ *vs.* $I_{DS}$) and the corresponding transconductance ($g_m$) of a functional OECT-unit in phosphate-buffered saline (PBS) were firstly acquired (also see *Figure 2—figure supplement 2*). The OECT-array was then characterized for its capability to detect the release of CA-NTs. Specific biological molecules were manually added to the electrolyte environment to mimic a pulsed release. In solutions (e.g., PBS) that are clear of any active background molecules, the detection limit can reach as low as 1 nM for pulsed release of dopamine, and was relatively consistent for different types of CA-NTs, including noradrenaline and adrenaline (*Figure 2—figure supplement 2*). In the artificial cerebral spinal fluid (ACSF) solution containing a high level of interfering background molecules (e.g., 1.28 mM ascorbate) that mimics the brain extracellular environment (*Mo and Ogorevc, 2001*), the OECT-array still showed obvious responses to the fluctuation of CA-NT concentration at 30 nM (*Figure 2c*). For quantitative analysis, the recorded $I_{DS}$ fluctuation was converted to the change of the effective GATE voltage ($\Delta V_{g-eff}$) using the I-V transfer characterization for each independent device. The $\Delta V_{g-eff}$ was further calibrated to the fluctuation of dopamine concentration in ACSF over a wide range of concentration values (30 nM ~ 0.1 mM), suggesting an extremely good dynamic range of the OECT-array for detecting CA-NTs, and the data could be fitted well with *Equation (2)* (*Figure 2d*).

## *In vivo* detection of CA-NTs using the OECT-array

We next examined the feasibility of using the OECT-array to detect the release of NTs, especially dopamine, in the brain of a living animal. An OECT-array was implanted to the VTA (−5.6 mm A/P, 0.8 mm M/L, 8 mm below dura) of an anesthetized rat using a stereotactic apparatus (*Figure 3a*). This region was selected because it is one of the main regions involved in dopaminergic signalling in a mammalian brain, mediating reward- and reinforcement-related behaviors (*Fields et al., 2007*). In parallel, a tungsten electrode was inserted to the medial forebrain bundle (MFB; −1.8 mm A/P, 2 mm M/L, 8 mm below dura) to evoke somatodendritic release of dopamine in VTA (*Kita et al., 2009*). Another tungsten electrode were bundled with the OECT-array for confirmative electrical recording (*Figure 3b*). Proper positioning of the OECT-array in VTA was confirmed by the spontaneous firing pattern. After being implanted to a rat brain, the $V_{GS}$-$I_{DS}$ transfer curve of the OECT-array was firstly acquired, and the derived transconductance curve was found to overlap well with the characterization acquired in ACSF (*Figure 3c*, also see *Figure 3—figure supplement 1*), suggesting

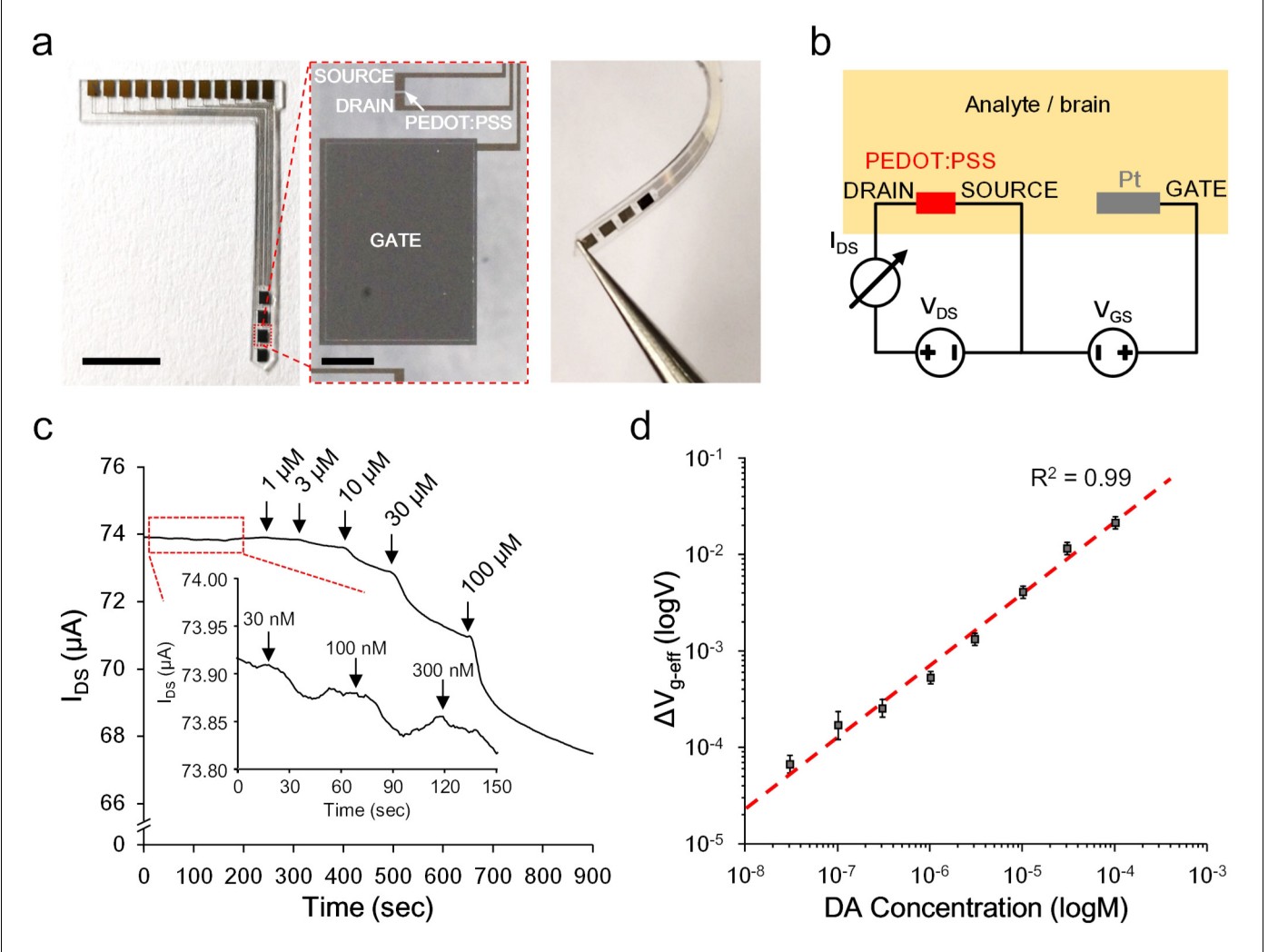

**Figure 2.** *Ex vivo* characterization of the OECT-array. (**a**) Photographs showing the overall (left), enlarged (middle) and bent view of the flexible OECT-array. Scale bar indicates 5 mm in left panel and 200 μm in the middle panel. (**b**) Wiring diagram showing the working setup of an OECT-array. (**c**) *Ex vivo* recording of $I_{DS}$ changes in response to artificially added dopamine to ACSF containing a high level of ascorbate acid (1.28 mM; $V_{DS}$ = 0.06 V; $V_{GS}$ = 0.6 V). (**d**) The calibration curve showing the relationship between the measured $\Delta V_{g\text{-eff}}$ and changes of dopamine concentrations, n = 3, error bars indicate standard error. The dash line shows linear fitting of the data.

The online version of this article includes the following source data and figure supplement(s) for figure 2:

**Source data 1.** Data for ex vivo device characterization.
**Figure supplement 1.** The dimension of the OECT-array.
**Figure supplement 2.** *Ex vivo* characterization of the OECT-array.

a stable device performance in brain tissues and the validity of the *ex vivo* calibration results (*Figure 2d*). Upon electrical stimulation (200 μA, 2 ms pulse width) in the MFB, the $I_{DS}$ of the 1st OECT-unit immediately showed a downward fluctuation, indicating the successful detection of a transient dopamine release. The release intensity was observed to linearly associate with the number of electrical pulses delivered to MFB (*Figure 3d,e*). When the stimulation sequence ramped from 1 to 100 electrical pulses, the amplitude of dopamine release in the VTA (as a result of transient release) rose from 36.13 ± 11.95 to 356.05 ± 52.07 nM (*Figure 3e*). When the stimulation electrode was moved out of the MFB to an irrelevant region (−1.8 mm A/P, 2 mm M/L, 3.5 mm below dura), even a very strong stimulation failed to induce any dopamine release. These results proved the basic feasibility of using the OECT for reliable and real-time detection of NT release in a mammalian brain.

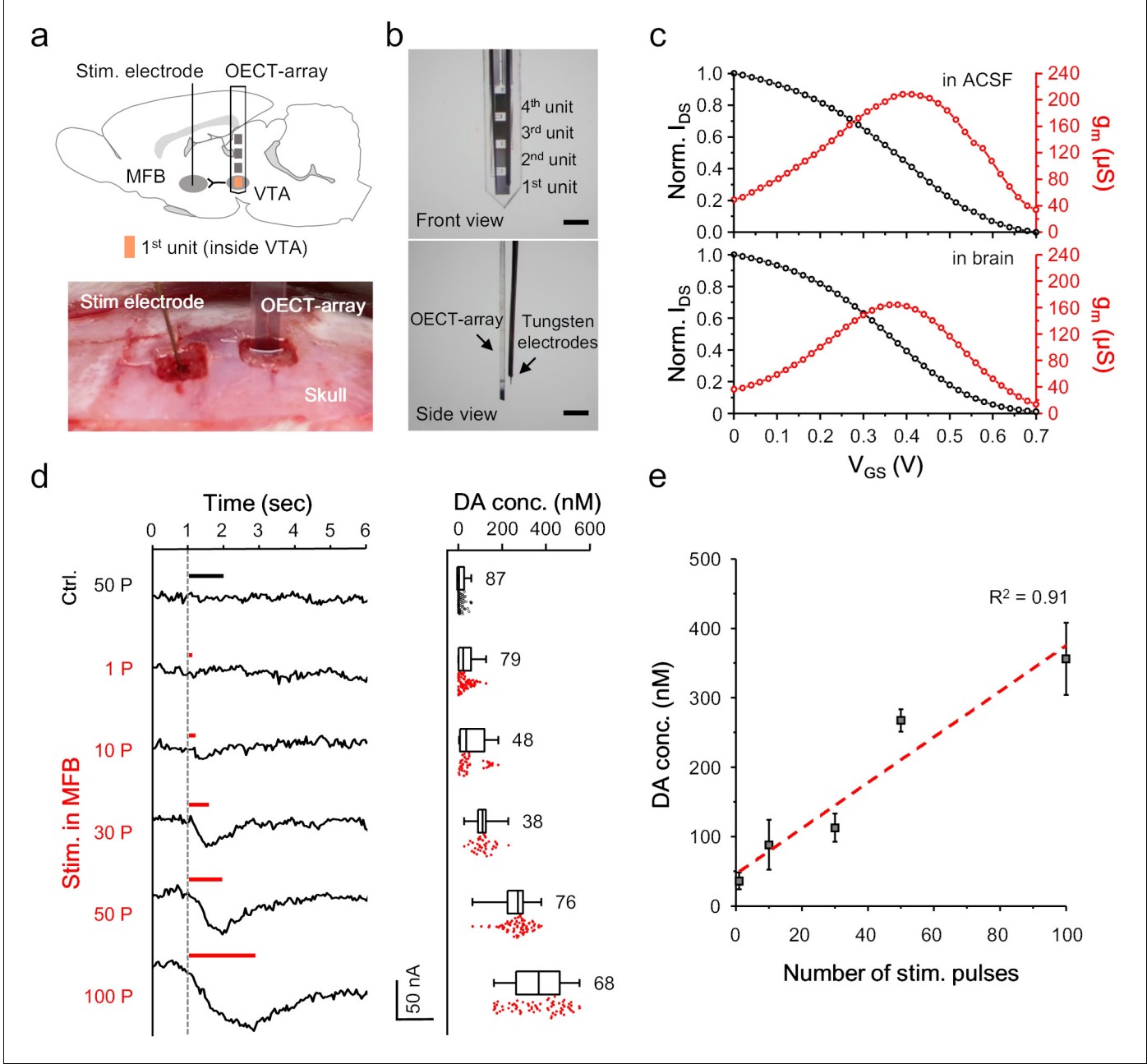

**Figure 3.** OECT-array for monitoring in VTA. (a) Schematic diagram (upper) and surgical photogram (lower) of the experimental setup. The 1st unit (orange) of an OECT-array was implanted to VTA to monitor somatodendritic dopamine release evoked by electrical stimulation in MFB. (b) Front (upper) and side (lower) view of the device bundle containing an OECT-array and a recording tungsten electrode. Scale bar, 1 mm. (c) Representative transfer (black) and transconductance (red) curve of an OECT device placed *ex vivo* in the ACSF (upper; $V_{DS}$ = 0.06V; $V_{GS}$ = 0.6V) or *in vivo* in a rat brain (lower; $V_{DS}$ = 0.06V; $V_{GS}$ = 0.6V). (d) The $I_{DS}$ recording (left) from an OECT-unit in the VTA in response to neural stimulation in MFB using different number of electrical pulses. The corresponding measurements of dopamine release from multiple trials were shown in the right boxplot. The whisker range is 1 ~ 99%, and each box show 25, 50% and 75% percentile of the data collected from three animals. For the control experiments, the stimulation was made off the MFB. (e) The correlation between the intensity of dopamine release in VTA and the number of electrical pulses in MFB, n = 3, error bars indicate standard error. The dash line shows linear fitting of the data.

The online version of this article includes the following source data and figure supplement(s) for figure 3:

**Source data 1.** Data for OECT recording in VTA.

**Figure supplement 1.** Characterization of OECT-array.

In another scenario, NTs are released at axonal terminals originated from the far away somata for signal transmission. We then examined the possibility of using the OECT-array to monitor dopamine release at remote axonal terminals after a long-range projection. Specifically, we focused on the mesolimbic pathway, in which the projection of dopaminergic neurons connects VTA and NAc (*Garris et al., 1999*). Accordingly, the OECT-array was implanted in the NAc (1.2 mm A/P, 1.4 mm M/L, 8.4 mm below dura), and a tungsten electrode was placed in the VTA for neural stimulation (*Figure 4a*, also see *Figure 4—figure supplement 1*). The surgical tracks in these two regions indicated the precise placement of the devices. Activation of relevant dopamine neurons was later confirmed by immunostaining for tyrosine hydroxylase (TH) and c-Fos in these cells (*Figure 4b*). In VTA, the majority of TH$^+$ staining was observed to be on neuron somata; however, in NAc, the TH$^+$ staining was mostly punctuated, suggesting an enrichment of dopaminergic axonal terminals in this region (*Figure 4b*). Before any electrochemical measurement, the polysynaptic connection between these two regions was firstly verified by electrical recording, which showed a clear temporal synchronization of spiking activity for neurons in the VTA and the NAc (*Figure 4—figure supplement 2*). To evoke dopamine release along the mesolimbic pathway, a 200 μA electrical stimulus (50 Hz, 2 ms pulse width, 50 pulses) was applied in the VTA to activate the dopaminergic neurons. Concurrently, the signals were transmitted to the NAc and monitored by the OECT-array. From the 1$^{st}$ OECT-unit placed in the NAc, a significant downward fluctuation of the $I_{DS}$ was recorded by the OECT-array (*Figure 4c*), and accordingly the evoked dopamine release was significantly higher than that of the control groups, in which the placement of the stimulation electrode and the recording OECT-unit were not paired on the dopaminergic pathway (*Figure 4d*). Notably, we found that the evoked dopamine release was frequency dependent, and a 50 Hz stimulation in the VTA induced the most significant changes (*Figure 4e*), which echoes a previous study that reported varying efficiency to evoke dopamine release by stimulating VTA using electrical signals of different frequencies (*Addy et al., 2010*). We then compared simultaneous recording from multiple OECT-units. As the placement of unit two was already away from the center of NAc, the recorded dopamine release was significantly lower than that from unit 1 (*Figure 4f*). These results demonstrated sensitive monitoring of NT release by the OECT-array at the remote axon projection terminals, and showed a potential for mapping neuronal electrochemical events by the array of multiple OECT-units on a single device.

## Mapping across different dopaminergic pathways

Activation of VTA/SNc complex simultaneously evokes dopamine release in NAc and CPu via mesolimbic and nigrostriatal pathways, respectively (*Garris and Wightman, 1994*). Our OECT-array provides the opportunity to simultaneously characterize the dopamine releasing profile across the two pathways, which cannot be precisely accessed by existing methods (*Schwerdt et al., 2017*). For each OECT-unit, the size of GATE electrode was 0.48 mm$^2$ and four of them (on one OECT-array) were sufficient to cover brain locations spanning from NAc to different parts of CPu (*Garris et al., 1999*; *Phillips et al., 2003*). In this experiment, the OECT-array was inserted into the striatum with an angle of 14° between the device and the dorsal-ventral axis. In this way, the 1$^{st}$ OECT-unit was placed in the NAc region, and the 3$^{rd}$ and 4$^{th}$ OECT units were placed in the lower and upper part of CPu (*Figure 5a*, specific coordinates provided in Materials and methods). As the 2$^{nd}$ OECT-unit was in the transition region between NAc and CPu, signals were not taken from this unit.

To specifically activate the mesolimbic pathway or nigrostriatal pathway, we delivered electrical stimulation to either VTA (−5.3 mm A/P, 0.8 mm M/L, 7.9 ~ 8.4 mm below dura) or SNc (−5.3 mm A/P, 2 mm M/L, 7.5 ~ 7.9 mm below dura) (*Figure 5b*), and investigated the corresponding dopamine release pattern in the striatum. We found that either VTA- or SNc-stimulation could evoke significant dopamine release in the large striatal region covered by the OECT-array (*Figure 5c*), which was not observed if the electrical stimulation was delivered to an irrelevant site (−5.3 mm A/P, 0.8 mm M/L, 4.9 mm below dura) (*Garris and Wightman, 1994*). Switching the stimulation site from VTA to SNc only affected the response in the NAc, but not in other striatal regions, suggesting a primary involvement of NAc in the mesolimbic pathway. However, it is interesting to find that SNc-stimulation also induced some dopamine release in the NAc, suggesting a possible cross-talk between the mesolimbic and the nigrostriatal pathways. Such connection between the two pathways was further evidenced by a similar response in the CPu (recorded by unit 3 and 4) to VTA- or SNc-stimulation.

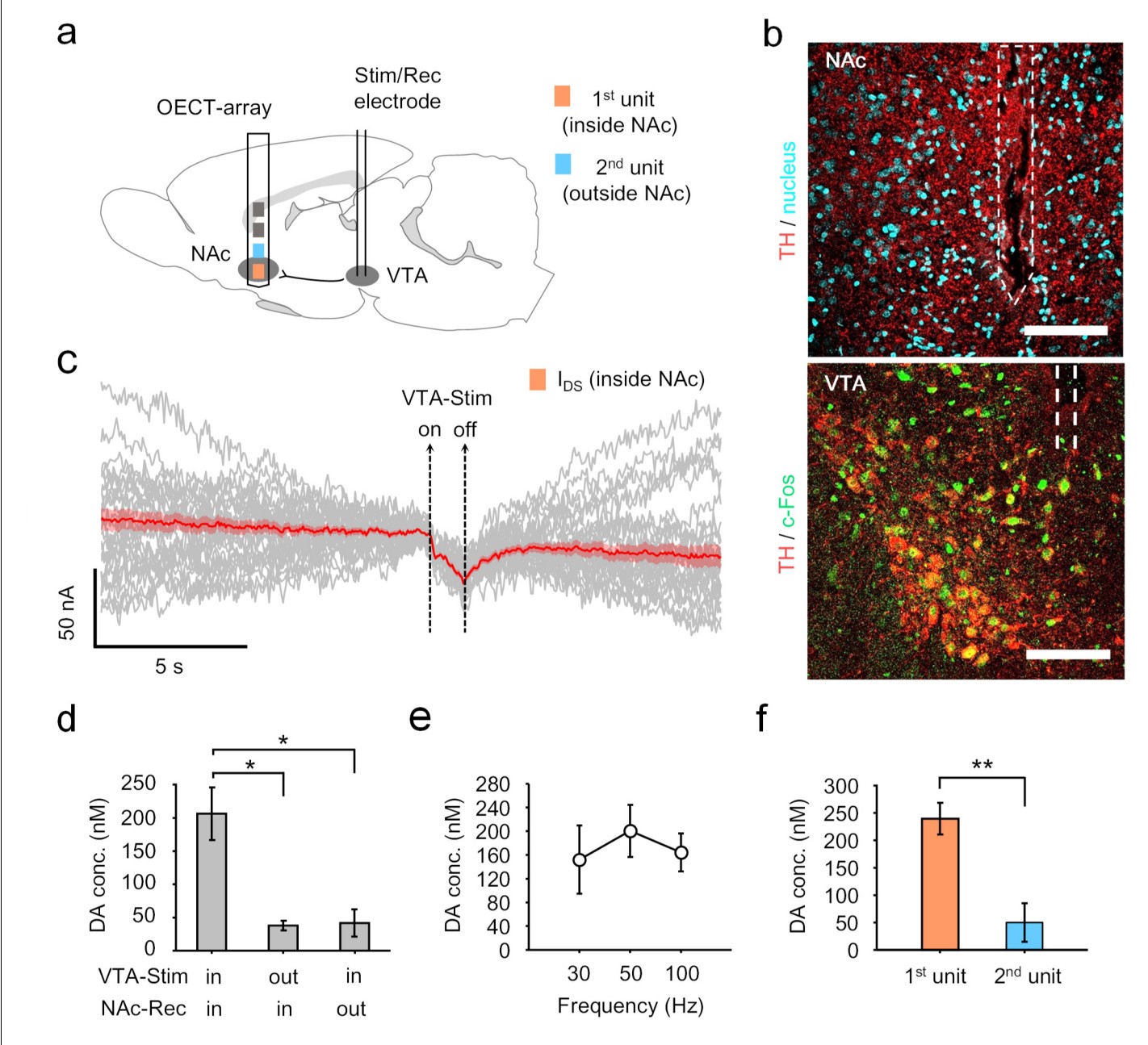

**Figure 4.** OECT-array for mapping dopamine release along the mesolimbic pathway. (**a**) Schematic diagram of using an OECT-array to map dopamine release around NAc region in response to neural stimulation of VTA. (**b**) Immunostaining of brain tissues in NAc (upper) or VTA (lower) after the electrochemical measurements. The NAc brain slices were stained for dopaminergic axon terminals (TH[+], red) and cell nuclei (DAPI, cyan); the VTA brain slices were stained for dopaminergic neurons (TH[+], red) and neural activation marker (c-Fos[+], green). The implantation track of the OECT-array (in NAc) and the stimulation electrode (in VTA) were indicated by the dashed lines. Scale bar, 100 μm. (**c**) The $I_{DS}$ recorded from the OECT-unit placed in NAc in response to electrical stimulation of VTA (2 ms pulse width, 50 Hz, 50 pulses; $V_{DS}$ = 0.05V; $V_{GS}$ = 0.65V). The gray curves are the raw $I_{DS}$ recording from multiple VTA-stimulation trials, the solid red curve shows the average of all 34 trials, and the red shade indicates the range of standard error. (**d**) Quantitative measurements of dopamine release in NAc in response to VTA-stimulation, n = 3, * indicates p<0.05 by student t-test. (**e**) Frequency-dependent dopamine release in the NAc as evoked by VTA stimulation (2 ms pulse width, 1 s duration, 30, 50, or 100 Hz), n = 3. (**f**) Comparison of dopamine release simultaneously measured by the 1st (inside NAc) and the 2nd (outside NAc) OECT-unit in response to VTA-stimulation (2 ms pulse width, 50 Hz, 1 s duration), n = 8, ** indicates p<0.005 by student t-test. For panel **d-f**, error bars indicate standard error.

The online version of this article includes the following source data and figure supplement(s) for figure 4:

**Source data 1.** Data for OECT mapping along the mesolimbic pathway.

**Figure supplement 1.** Confirmation of correct OECT-unit placement in NAc.

*Figure 4 continued on next page*

*Figure 4 continued*

**Figure supplement 2.** Electrophysiological characterization in the NAc in response to VTA-stimulation.

Based on the dopamine mapping in the striatum, we then hypothesized a mutual innervation mechanism between VTA and SNc that contributes to the cross-talk between the mesolimbic and the nigrostriatal pathways. To validate this hypothesis, we firstly conducted electrophysiology recording to probe dopamine neuron activity in the SNc in response to VTA-stimulation or vice versa, and confirmed a reciprocal excitation between these two nuclei (*Figure 6—figure supplement 1*). Then, we used a blade (~200 μm thick,~1.3 mm wide) to make a mechanical lesion between the VTA and the SNc, which physically disconnected the two regions (*Figure 6a,b*). Such disruption has differential effects on dopamine release in different parts of striatum. In the NAc, cutting off VTA-SNc connections lowered the dopamine release in response to VTA-activation, and substantially reduced the response under SNc-stimulation (*Figure 6c*), suggesting that SNc is involved in meso-limbic signalling via the reciprocal connection between VTA and SNc. Similarly, in the lower part of CPu, we observed a reduction of dopamine release under either VTA- or SNc-stimulation after the physical lesion (*Figure 6d*), suggesting that VTA also partially affects nigrostriatal pathway through the mutual innervation between VTA and SNc. However, in the upper part of CPu, we found that

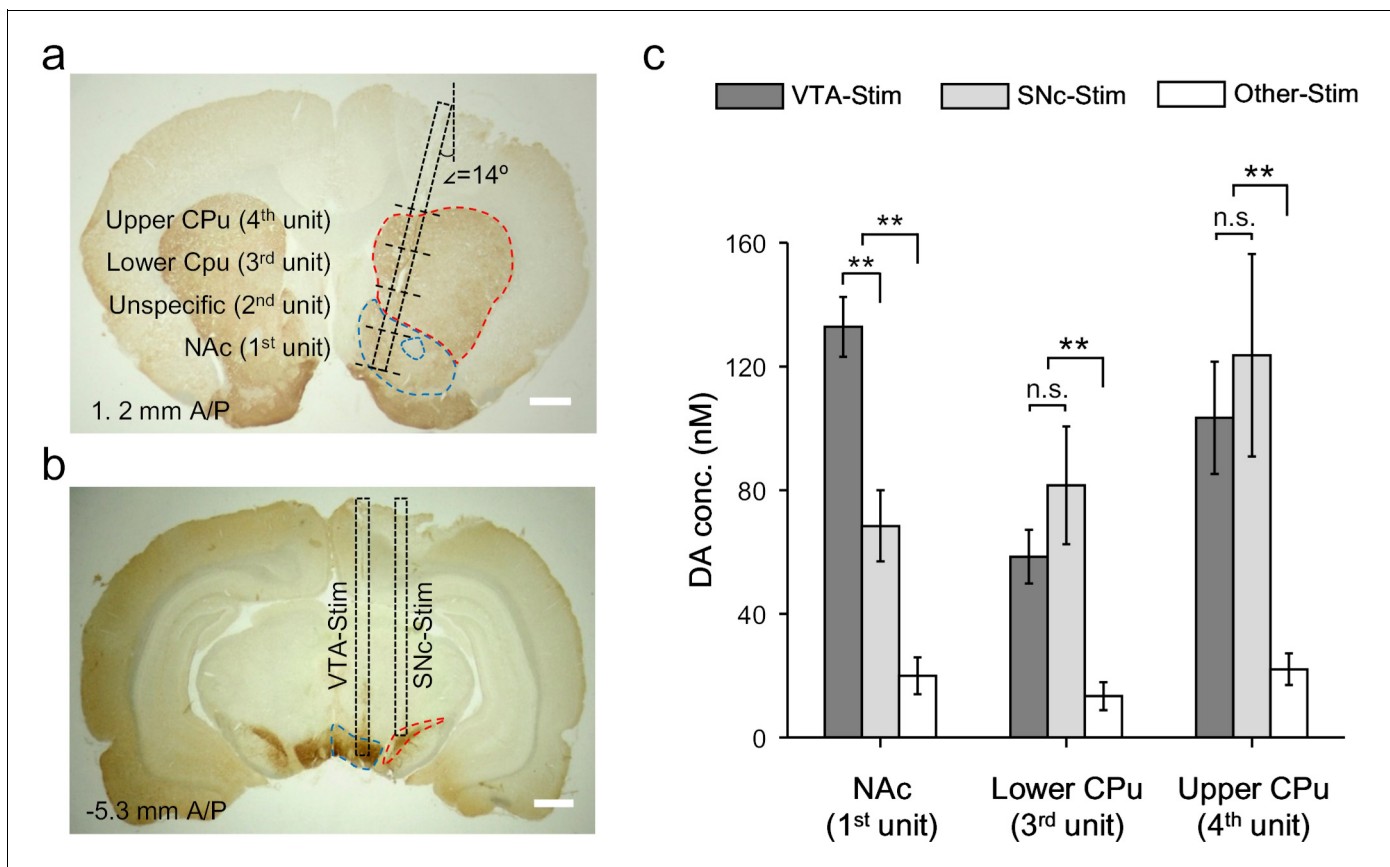

**Figure 5.** Dopamine mapping across the mesolimbic and nigrostriatal pathways. (**a**) Immunohistochemical staining for TH in the CPu and the NAc after an electrochemical measurement. The surgical track of the OECT-array was indicated by the dashed box. The location of each OECT-unit was denoted by dashed lines. Scale bar, 1 mm. (**b**) Immunohistochemical staining for TH in the VTA and SNc. The surgerical tracks of the stimulation electrodes in VTA or SNc were indicated by dashed boxes. Scale bar, 1 mm. (**c**) Quantitative analysis of dopamine release in NAc and different parts of CPu simultaneously measured by multiple OECT-units upon electrical stimulation of the mesolimbic (in VTA) or the nigrostriatal (in SNc) pathways. n = 10, the error bars indicate error, ** indicates p<0.005 by student t-test, n.s. stands for 'not significant'.

The online version of this article includes the following source data for figure 5:

**Source data 1.** Data for multi-region recording of dopamine release.

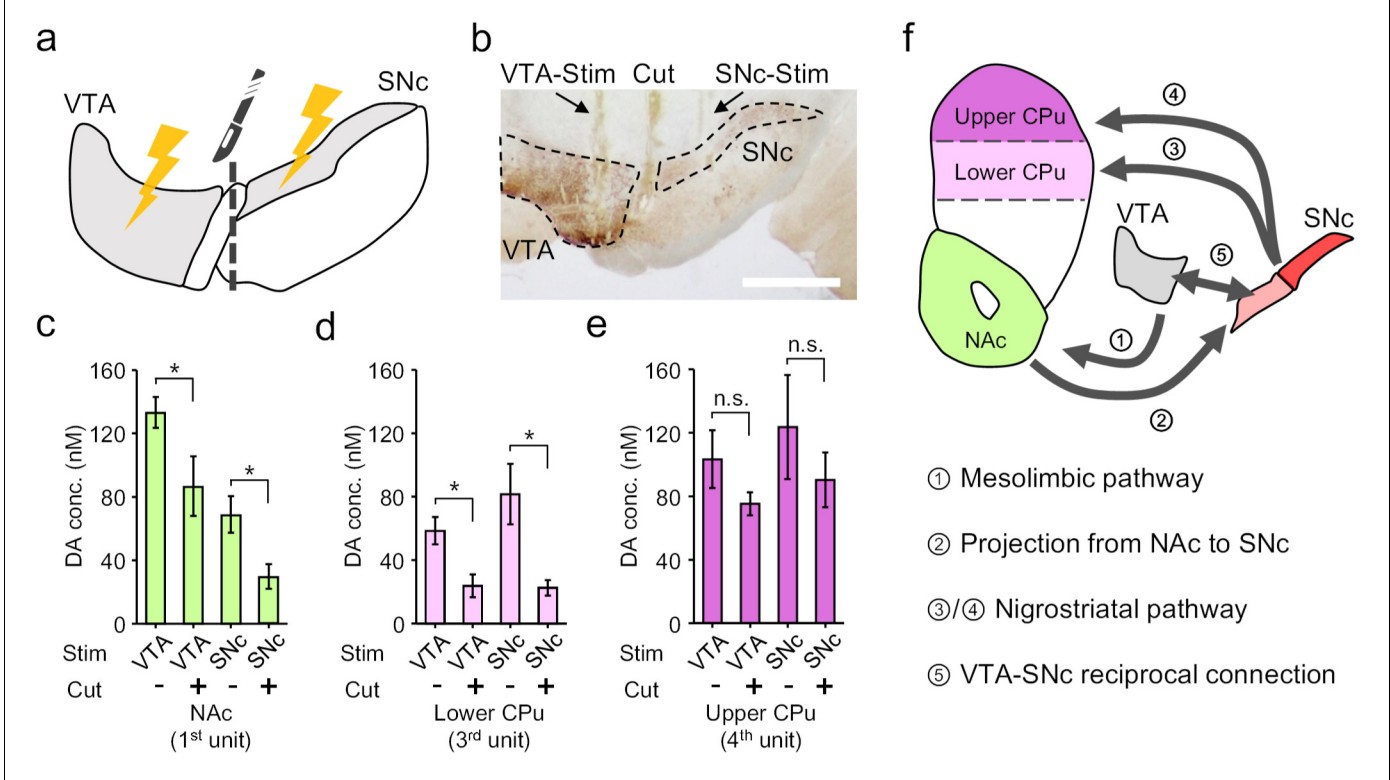

**Figure 6.** Electrochemical cross-talk between the mesolimbic and nigrostriatal signalling. (a) Schematic diagram of dopamine mapping in the NAc and CPu in response to surgically isolated VTA- or SNc-stimulation. (b) Immunohistochemical staining for TH in a brain slice showing the surgical tracks of the stimulation electrode in the VTA or SNc, and the mechanical lesion between these two regions. Scale bar, 1 mm. (c–e) Quantitative analysis of the change in dopamine release pattern at different brain regions across mesolimbic and nigrostriatal pathways, including the NAc (c), lower CPu (d) and upper CPu (e), in response to VTA- or SNc-stimulation before and after the surgical lesion to mechanically break the mutual connections between VTA and SNc. n = 6, error bars indicate standard error, *indicates p<0.05 by student t-test; n.s. stands for 'not significant'. +/- indicates that the measurements were conducted with/without the VTA-SNc surgical lesion. (f) Summary of the identified complex cross-talk between the mesolimbic and nigrostriatal pathways as regulated by the mutual connection between VTA and SNc.

The online version of this article includes the following source data and figure supplement(s) for figure 6:

**Source data 1.** Data for change in dopamine release after VTA-SNc surgical lesion.

**Figure supplement 1.** Reciprocal connection between VTA and SNc characterized by electrophysiology.

dissociation of VTA and SNc had almost no effects on either VTA- or SNc-stimulation induced response (*Figure 6e*). This observation indicated that the VTA-SNc reciprocal connection is not involved in the nigrostriatal signalling from SNc to upper CPu. However, signals resulted from the activation of VTA can still reach upper CPu to induce neurochemical response (*Figure 6e*), which is a sign for the existence of a cascade transmission circuit from VTA to upper CPu (*Pennartz et al., 2009*; *Figure 6f*). Taken together, the mapping of the electrochemical events by using the OECT-array experimentally demonstrated a cross-talk between the mesolimbic and nigrostriatal dopaminergic pathways, and also supported a heterogeneous projection from SNc to CPu, which is differentially affected by VTA-SNc reciprocal connections.

## Discussion

Here, we describe an OECT-array technique for real-time and multi-site monitoring of electrochemical release of CA-NTs in brains of living animals. The device is fabricated from an organic semiconductive PEDOT:PSS layer (*Mo and Ogorevc, 2001*), which is integrated with traditional electronic components on a flexible membrane, and configured into a highly sensitive transistor-based array for biosensing applications. As we have demonstrated in this study, the OECT-array is an independent recording system, which does not require a reference electrode as in the traditional CV

(*Roberts and Sombers, 2018*; *Robinson et al., 2003*). The dimensions of each OECT-unit, especially the GATE electrode size, and whole OECT-array were carefully designed based on the stereotaxic coordinates, which covers the striatal brain region and maximizes the recorded NT release signal. Such simple configuration enables each OECT-unit to act as a fully functional sensor, and leads to a greatly easier implantation process and a more convenient recording protocol.

Compared to the passive electrodes (e.g., carbon fiber) that are widely used in CV for *in vivo* electrochemical characterization, the OECT-array is an active sensor with intrinsic amplification capability during signal recording, which is a key feature to eliminate the noise accumulated from surrounding environment over the amplification process (*Rivnay et al., 2017*). Unlike many CV-based measurements, in which signal is usually overwhelmed by noise due to the high-speed voltage scanning, the typical signal-to-noise ratio (SNR) of the OECT-device ranges from 5 to 10 (depending on different recording sites in a brain), given an *in vivo* peak-to-peak noise level of $1.16 \pm 1.52$ nA (n = 26). This feature enables real-time and direct data readout without the need of afterwards background subtraction or complex mathematical fitting as required in traditional CV experiments (*Bucher and Wightman, 2015*). As another key advantage, the working voltage of the OECT-array is only ~300 mV, which is almost 80% reduced from the required voltage in CV measurements. Such a low operating voltage causes lower power consumption and also induces less tissue electrolysis during *in vivo* recording (*Bikson et al., 2009*), which is critical for long-term recordings. As a result, the OECT-array can be used to continuously monitor the electrochemical activity in a brain for hours, which is inaccessible for carbon-fiber measurement that normally continues for minutes because of significant signal drift (*Roberts and Sombers, 2018*). Even compared to the genetic approach that was recently reported to measure dopamine release *in vivo* (*Patriarchi et al., 2018*), the OECT-array is still advantageous in its quantification capability, ease-of-use, and flexibility for detecting different NT species, and great potential for interference-free behavioural experiments.

The OECT sensor achieves a detection limit of 30 nM for dopamine in ACSF (containing 1.28 mM ascorbate), demonstrating the capability for dopamine detection at low concentration with the existence of high level interferences. Considering that the concentration of ascorbate *in vivo* is homeostatic while evoked dopamine signals perform fast phasic change, we believe that such high level of electrochemical active interferences does not affect the detection of dopamine *in vivo* (*Phillips et al., 2003*; *Garris et al., 1999*). In this study, the sensor was used to detect dopamine release in brain regions with dominant dopaminergic signalling upon neural stimulation. The Pt-GATE electrode shows a strong electro-oxidation reaction with released dopamine, which can distinguish dopamine from other interference NTs in the brain environment (e.g., glutamate and GABA; *Figure 2c*, also see *Figure 2—figure supplement 2*). Though 3,4-dihydroxyphenylacetic acid (DOPAC) has catechol that can be electro-oxidized on Pt, its conversion from dopamine requires monoamine oxidase, and operates in the tonic manner (on minute timescale), which would not affect the sensing of phasic dopamine overflow (on millisecond timescale) using the OECT-array (*Phillips et al., 2003*). With further development, the selectivity of the OECT device could be improved by different strategies. For example, surface of the Pt-GATE electrode can be modified with a layer of biocompatible polymer, Nafion (*Liao et al., 2015*), which is a polyelectrolyte with stable Telfon backbone and acidic sulfonic groups, and is negatively charged in neutral solutions. Also, the selectivity could be customized through a multilayer (Nafion/PANI/enzyme/graphene) modification of the Pt GATE (*Liao et al., 2015*). A particular NT could be selectively detected by choosing the coating enzyme (e.g., dopamine β-hydroxylase) within the multilayer sandwich. Some of these strategies have been demonstrated *in vitro* (*Liao et al., 2015*), and can be further optimized for *in vivo* applications.

While OECT-based device has been previously used to detect neuronal action potentials (*Khodagholy et al., 2013*), this study is first to implement an electrochemical sensor for *in vivo* applications. The differentiation between evoked electrical activity and neurotransmitter release is determined by the configuration of the OECT device. The performance and nature of an OECT are dominated by the ratio of the capacitance of the GATE/electrolyte ($C_{G-E}$) and that of the PEDOT: PSS channel layer ($C_{volumetric}$) (*Figure 1c*), which is reversely proportional to the ratio of potential drop at the two interfaces on a OECT device (*Yaghmazadeh et al., 2011*). For device used to record electrical activity, the GATE/electrolyte capacitance is larger than the capacitance of channel, leading to a larger potential at the channel layer when the total voltage applied across GATE/electrolyte/channel is maintained constant. Change of ion concentration upon neuronal spiking activity

influences the amount of cations injected into the channel (driven by the GATE voltage) and affects the conductivity through doping and de-doping the channel, which favors the detection of transient electrical signals (*Khodagholy et al., 2013*). For electrochemical sensing, the OECT-device is more complicated because of a trade-off between chemical detection and effective gating of the transistor. When faradic current is generated at the GATE/electrolyte interface as a result of electrochemical reactions, the GATE/electrolyte potential decreases, the channel/electrolyte potential increases, and the channel is de-doped. Subsequently, the channel conductivity is lowered and the current flowing through channel is decreased. In principle, chemical detection on the Pt-GATE demands a smaller GATE/electrolyte capacitance (compared to the channel/electrolyte capacitance) to enhance detection sensitivity, whereas effective gating requires vice versa. However, the GATE electrode could not be miniaturized too much, in order to maintain certain potential drop on the channel and to avoid a weak GATE modulation of the device. It is therefore important to tune the capacitance ratio more carefully in this case. The capacitance of Pt-GATE electrode for detection of catecholamine is an electrical double layer (EDL), which is estimated to be around 4.8 nF (10 $\mu$F/cm$^2$, $0.08 \times 0.06$ cm$^2$), considering the EDL of Pt per unit area and the electrode size (*Drüschler et al., 2010*). By tuning the size of the channel, the volumetric capacitance of PEDOT:PSS channel in electrolyte is estimated to be around 0.47 nF (39 F/cm$^3$, $0.03 \times 10 \times 40$ $\mu$m$^3$), which is effective for gating to amplify signal (*Rivnay et al., 2015*), and the oxidation of catecholamine on the GATE electrode is maximized while gating is kept effective, so as to configure the OECT as chemical sensor in favor of electrochemical detection.

Using the OECT-based technology, we demonstrated successful detection of evoked dopamine release in different physiological scenarios. Notably, the microarrayed format of the device further enabled a spatial mapping of dopamine release simultaneously at multiple sites over a large brain region. Taking the dopaminergic pathways as a working model, our results from the OECT-array facilitated electrochemical mapping reveal a complex cross-talk between the mesolimbic and nigrostriatal signalling, which relies on a reciprocal innervation between VTA and SNc. These two nuclei are frequently treated as a whole in many studies, though their cell compositions are dramatically different (*Garris and Wightman, 1994*). Actually, the VTA/SNc mutual connections has also been shown by other studies. For example, CLARITY with highly cell-specific tracing revealed that SNc-neurons projecting to dorsomedial and dorsolateral striatum receive a small portion of neural input from VTA (*Lerner et al., 2015*). However, in this study, we cannot rule out the possibility that other indirect network or multi-synaptic transmission (*Watabe-Uchida et al., 2012*) contributed to results primarily made by the electrical stimulation and OECT recording. In addition, it has recently been reported that NAc not only receives axonal projection from VTA but also projects back to VTA as well as to SNc (*Belin and Everitt, 2008*). Therefore, it is not surprising to see that activation of VTA can induce significant dopamine release in different parts of CPu even after a mechanical lesion of the VTA-SNc connection, probably through an indirect circuit involving the feedback projections (*Figure 6f*). Yet, it is intriguing to observe a different electrochemical response in the upper and lower part of CPu as a result of the lesion, supporting the existence of distinct functional divisions within SNc, which could be heterogeneously associated with VTA via reciprocal innervations. These observations are well in line with the recent report about differential circuit incorporation of subpopulations of SNc dopamine neurons that project to distinct regions of dorsal striatum (*Lerner et al., 2015*; *Keiflin and Janak, 2015*), and hint the possibility of their specific links between particular subcategory of dopamine neurons in VTA (*Bourdy et al., 2014*; *Yetnikoff et al., 2014*), which can be further explored with an array of circuit interrogation tools in more details.

In summary, this proof-of-concept study demonstrates the development and application of the OECT-array as a powerful and easy-to-use platform for electrochemical analysis in the nervous system. From an instrumentation perspective, the OECT-array can be highly configurable and expandable for different experimental scenarios. For example, a similar OECT-based device can be forged into a flexible planar format for mapping electrocorticography signals over a large brain surface (e. g., cortex) (*Khodagholy et al., 2013*; *Lee et al., 2017*); more sensing units can be easily integrated to achieve higher spatial resolution; and different organic semiconducting materials can also be tested for the development of more specific and sensitive transistor-based electrochemical sensors (*Rivnay et al., 2018*; *Venkatraman et al., 2018*; *Pappa et al., 2018*). We believe that the OECT-array can be a readily applicable platform to supplement existing toolsets for brain circuit interrogation. Combined with other neuro-techniques, such as electrophysiology, optogenetics, viral

tracing, etc., it can potentially lead to a wide range of novel neuroscience research that fully utilizes the electrical, optical, and biochemical information from a living animal brain.

## Materials and methods

**Key resources table**

| Reagent type (species) or resource | Designation | Source or reference | Identifiers | Additional information |
|---|---|---|---|---|
| Strain, strain background | Sprague Dawley rat | Chinese University of Hong Kong | | |
| Antibody | rabbit anti-tyrosine hydroxylase | Abcam | Cat# ab112 | IF(1:1000) |
| Antibody | sheep anti-c-Fos antibody | Abcam | Cat# ab6167 | IF(1:1000) |
| Chemical compound, drug | AZ 5214 photoresist | MicroChem | | |
| Chemical compound, drug | AZ 400K developer | MicroChem | | |
| Chemical compound, drug | PEDOT:PSS | Heraeus | Clevios PH-500 | |
| Chemical compound, drug | GOPS | International Laboratory | Cat# 1094601 | |
| Software, algorithm | Matlab | Matlab | 2014a/2015a | |

### Fabrication of the OECT-array

The fabrication of the OECT-array mainly involves sequential deposition and patterning of the gold (Au)/chromium (Cr) SOURCE and DRAIN electrodes, the Pt-GATE electrode, the PEDOT:PSS (Clevios PH-500, Heraeus) channel active layer, and the SU-8 (MicroChem) insulating layer (*Wang et al., 2018*). First, the 200 μm thick polyethylene terephthalate (PET) substrates were cleaned by ultrasonication in chemical solvent (acetone and isopropanol; Sigma-Aldrich) and oxygen plasma (Harrick Plasma) treatment. Then, the Au (100 nm)/Cr (10 nm) electrodes were deposited on the PET substrate by magnetron sputtering and patterned through a lift-off process. Specifically, a layer of AZ5214 photoresist (MicroChem) was spin-coated on the PET substrate, patterned by UV exposure, and then developed using AZ 400K developer (MicroChem). The Au/Cr electrodes was deposited and patterned by a lift-off process, forming the SOURCE and the DRAIN electrode, which were spaced by a channel of 10 μm in length and 40 μm in width. Following the same procedure, a layer of Pt (100 nm)/titanium (10 nm) was deposited to form the Pt-GATE measured at $600 \times 800$ μm$^2$. The PEDOT:PSS aqueous dispersion was first mixed with dimethyl sulfoxide (DMSO; 5% in volume ratio; Sigma-Aldrich), glycerin (5% in volume ratio; Sigma-Aldrich) and (3-glycidyloxypropyl) trimethoxysilane (GOPS; 1% in volume ratio; International Laboratory) to enhance the conductivity and film stability. Then the PEDOT:PSS was spin-coated and annealed at 150°C in nitrogen for 1 hr. Unwanted PEDOT:PSS was removed along with the photoresist by rinsing in acetone (Sigma-Aldrich). The device was packaged by patterning a layer of photoresist (2 μm; SU-8 2002) to insulate the metal electrodes from aqueous electrolyte. The devices were cleaned in PBS solution before use.

### Recording setup for an OECT-array

In an electrochemical analysis using the OECT-array, multiple dual-channel sourcemeters (2612/2614, Keithley) were used. The OECT-units were electrically separated from each other. To acquire the transfer curve for each OECT-unit, the $V_{DS}$ was maintained constant, while $V_{GS}$ was swept from 0 to 1.2 V for *ex vivo* (or *in vivo*) analysis. During signal recording, the applied $V_{GS}$ was determined by referencing to the transfer curve to maximize the transconductance ($g_m$) for the best signal amplification. For *ex vivo* measurements, the neurotransmitter molecules (e.g., dopamine, noradrenaline and adrenaline; Sigma-Aldrich) with designed concentrations were added to PBS solution to mimic a

pulsed release, and the fluctuation of $I_{DS}$ as a function of time was recorded. For *in vivo* measurements, the evoked $I_{DS}$ fluctuation was recorded upon electrical stimulation of the neuronal circuits in the brain of an anesthetized animal. All data was collected using the software TSP-express (Keithley).

## Animal surgery

All experimental procedures involving animals were approved by the university Animal Ethics Committee. Animal licenses, (16-97) in DH/HA and P/8/2/5 Pt.5 and (18-129) in DH/SHS/8/2/5 Pt.4, were approved by Department of Health of the Government of Hong Kong Special Administration Region. Both male and female Sprague Dawley rats (300 ~ 400 g) were used. Before the experiments, the animals were firstly anesthetized with intraperitoneal injection of urethane (2 g/kg; Sigma-Aldrich), and then mounted on a stereotactic frame (Narishige). A heating pad was placed underneath the animals to maintain a temperature of 37°C throughout an experiment. For monitoring somatodendritic dopamine release in VTA (*Figure 3a*), two holes ($2 \times 2$ mm$^2$) were drilled on the skull. The stereotactic coordinates for stimulation in MFB was −1.8 mm A/P, 2 mm M/L, 8 mm below dura. The stereotactic coordinates for the 1$^{st}$ OECT unit was −5.6 mm A/P, 0.8 mm M/L, 8 mm below dura. The control experiments were conducted in the same way except that the stimulation is moved out of MFB (−1.8 mm A/P, 2 mm M/L, 3.5 mm below dura). For monitoring dopamine release remotely along the mesolimbic pathway (*Figure 4a*), the tungsten electrode was implanted in the VTA (−5.6 mm A/P, 1 mm M/L, 7.9 ~ 8.4 mm below dura) for electrical stimulation, and the OECT-array was implanted in the NAc (1$^{st}$ OECT unit; 1.2 mm A/P, 1.4 mm M/L, 8.4 mm below dura) for electrochemical monitoring. In the control experiments, the OECT-array was not moved, and the electrical stimulation was delivered to an irrelevant region (−5.6 mm A/P, 1 mm M/L, 4.9 mm D/V below dura). For investigations about the cross-talk between the mesolimbic and the nigrostriatal pathway (*Figure 5a*), a tungsten electrode (FHC) was either implanted at VTA (−5.3 mm A/P, 0.8 mm M/L, 7.9 ~ 8.4 mm below dura) or SNc (−5.3 mm A/P, 2 mm M/L, 7.5 ~ 7.9 mm below dura) for electrical stimulation of dopaminergic signalling pathway. In the control experiments, the stimulation was delivered to a region out of VTA or SNc (−5.3 mm A/P, 0.8 mm M/L, 4.9 mm below dura). The OECT-array was inserted to an exposed rat brain (1.2 mm A/P, 3 mm M/L) with 14° tilted towards the lateral side, and was allowed to travel 7.2 ~ 7.4 mm. In this configuration, the 1$^{st}$ unit was placed in NAc and the coordinates were 1.2 mm A/P, 1.4 mm M/L, 7.1 mm below dura; the 3$^{rd}$ unit was placed lower CPu and the coordinates were 1.2 mm A/P, 1.9 mm M/L, 4.7 mm below dura; the 4$^{th}$ unit was placed in upper CPu and the coordinates are 1.2 mm A/P, 2.2 mm M/L, 3.5 mm below dura. After an experiment, the animal was sacrificed and transcardially perfused with chilled PBS and then 4% Paraformaldehyde (PFA; in PBS). The brain was isolated and stored in 4% PFA (in PBS) for later anatomical or immunohistochemical analysis.

## Electrical stimulation

To evoke action potentials in brain tissue, a tungsten metal electrode (~200 μm in diameter) was used to deliver current stimulation to appropriate brain locations. To evoke somatodendritic dopamine release (*Figure 3*), the electrical stimuluses (50 Hz, 2 ms pulse width, 200 μA amplitude; AMPI) of different duration (1, 10, 30, 50, and 100 pulses) were delivered in MFB. To detect the NT release at axonal projection terminals (*Figures 4–6*), a pulse sequence (50 Hz, 2 ms pulse width, 1 s duration, 200 μA amplitude) was used. The interval between each stimulation is at least 50 s to avoid NT depletion.

## Immunohistochemistry

An isolated brain was fixed in 4% PFA for 2 days followed by soaking in 30% sucrose until it settled to the bottom. The sample was then frozen-sectioned (Thermo Scientific) into coronal slices of 25 or 50 μm thickness for further processing. Before immunostaining, the brain slices were rehydrated in PBS for 2 hr and blocked by 3% bovine serum albumin (BSA) in tris-buffered saline with 0.25% Triton X-100 (TBST) for 2 hr. Next, the slides were incubated with primary antibodies at 4°C overnight followed by a thorough rinse in TBST solution. The slices were further incubated with secondary antibodies at room temperature for 2 hr and afterward DAPI solution (10 μg/ml; Abcam) for 30 min. After rinsing in TBST, the slides were dehydrated in 30% ethanol, 50% ethanol, 75% ethanol, 100% ethanol and 100% xylene sequentially, and mounted for further imaging and storage. Specifically for

the horseradish peroxidase/3,3'-Diaminobenzidine (HRP/DAB) staining, the sectioned brain slices were rehydrated in PBS for 2 hr and were blocked by hydrogen peroxide blocking solution (Abcam) for 30 min and by 3% BSA in TBST for 2 hr. Afterwards, the slices were incubated in primary antibody at 4°C overnight. After a thorough wash in TBST solution for 10 min for three times, the slices were further incubated in biotinylated secondary antibody (Abcam) for 30 min followed by TBST wash for 10 min for three times. Next, the peroxide-labeled streptavidin solution (Abcam) was applied for 30 min and subsequently washed in TBST for 10 min for three times. The DAB chromogen (Abcam) was diluted in DAB substrate (Abcam) to working concentration, and the diluted chromogen solution was applied directly to the slides. At satisfactory staining level, the slides were rinsed in deionized water to stop the development. The slices were dehydrated and mounted in the same way aforementioned for imaging and storage. The antibodies used in this study are: rabbit anti-tyrosine hydroxylase (0.3 μg/ml; ab112, Abcam) and sheep anti-c-Fos antibody (2 μg/ml; ab6167, Abcam).

## Microscopy

The fluorescently stained brain slides were imaged using a laser scanning confocal microscope equipped with a 40 × water immersion lens (SP8, Leica). For each slice, the scan was performed with 1 μm z-resolution, and a maximum projection of each scan was then acquired. For overall anatomical evaluation, the HRP/DAB-stained brain slides were imaged using a stereoscope.

## Statistical analysis

A MATLAB program was developed to convert the recorded change of $I_{DS}$ to $\Delta V_{g\text{-eff}}$ by using the transfer curve to quantitatively analyze the level of recorded NT release as recorded by individual OECT-unit. Specifically, the $I_{DS}$ before electrical stimulation was converted to voltage value according to the transfer curved (as baseline); the peak $I_{DS}$ (the minimal $I_{DS}$ value evoked by an electrical stimulation before reversing back to baseline) in response to NT release was also converted to voltage value in the same manner; these two voltage values were further subtracted and absolutized to obtain $\Delta V_{g\text{-eff}}$. The calculated $\Delta V_{g\text{-eff}}$ was further converted to change of molecular concentration by using a calibration curve fitted from the *ex vivo* experimental results. For statistical analysis, student t-test was performed to determine the statistical significance between the experimental conditions and the control groups, $p < 0.05$ indicates a significant difference. At least three independent biological replicates were used if not otherwise specified. For *Figure 3*, data from ~20 trials were collected from each animal. For *Figure 4*, ~30 trials were performed on each animal. For *Figure 5*, the data were collected from 10 animals, and ~30 trials were performed on each animal. For *Figure 6*, the data were collected from six animals, and ~30 trials were performed on each animal.

## Acknowledgements

This work was supported by General Research Fund (11218015, 11278616, 11203017) and Collaborative Research Funds (C5015-15G) from the Research Grants Council of Hong Kong SAR, and by Health and Medical Research Fund (06172336) from the Food and Health Bureau of Hong Kong SAR. We also thank funding support by the Science Technology and Innovation Committee of Shenzhen Municipality (JCYJ2017081810034239, JCYJ20180507181624871) and National Natural Science Foundation of China (81871452). Funds from City University of Hong Kong (7005084, 7005206) are also acknowledged.

## Additional information

### Funding

| Funder | Grant reference number | Author |
|---|---|---|
| Research Grants Council, University Grants Committee | 11278616 | Peng Shi |
| Research Grants Council, University Grants Committee | 11218015 | Peng Shi |

| | | |
|---|---|---|
| Research Grants Council, University Grants Committee | 11203017 | Peng Shi |
| Health and Medical Research Fund | 06172336 | Peng Shi |
| Research Grants Council, University Grants Committee | C5015-15G | Feng Yan<br>Peng Shi |
| Science Technology and Innovation Committee of Shenzhen Municipality | JCYJ20170818100342392 | Peng Shi |
| Science Technology and Innovation Committee of Shenzhen Municipality | JCYJ20180507181624871 | Peng Shi |
| National Natural Science Foundation of China | 81871452 | Peng Shi |
| City University of Hong Kong | 7005084 | Peng Shi |
| City University of Hong Kong | 7005206 | Peng Shi |

The funders provided resources for the study design, data collection, and interpretation.

### Author contributions

Kai Xie, Data curation, Software, Formal analysis, Investigation, Methodology, Writing - original draft; Naixiang Wang, Data curation, Formal analysis, Investigation, Writing - review and editing; Xudong Lin, Methodology; Zixun Wang, Peilin Fang, Data curation; Xi Zhao, Haibing Yue, Data curation, Methodology; Junhwi Kim, Validation; Jing Luo, Shaoyang Cui, Writing - review and editing; Feng Yan, Conceptualization, Resources, Supervision, Funding acquisition, Writing - review and editing; Peng Shi, Conceptualization, Resources, Supervision, Funding acquisition, Investigation, Writing - original draft, Project administration, Writing - review and editing

### Author ORCIDs

Kai Xie (iD) https://orcid.org/0000-0002-9090-8383
Peng Shi (iD) https://orcid.org/0000-0003-0629-4161

### Ethics

Animal experimentation: All experimental procedures involving animals were approved by the university Animal Ethics Committee. Animal licenses, (16-97) in DH/HA&P/8/2/5 Pt.5 and (18-129) in DH/SHS/8/2/5 Pt.4, were approved by Department of Health of the Government of Hong Kong Special Administration Region.

### Decision letter and Author response

Decision letter https://doi.org/10.7554/eLife.50345.sa1
Author response https://doi.org/10.7554/eLife.50345.sa2

## Additional files

### Supplementary files

- Source code 1. Matlab code for signal and data processing.
- Transparent reporting form

### Data availability

All data generated or analysed during this study are included in the manuscript and supporting files. Source data files have been provided as supplementary files.

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
