## [Decision Letter]

**Acceptance summary:**

This paper uses organic transistors to detect neuromodulators deep within the brain with nanomolar sensitivity and milliseconds resolution. The use of organic transistors also enhances scalability of such recording techniques to cover multiple brain areas.

**Decision letter after peer review:**

Thank you for submitting your article "Organic electrochemical transistor arrays for real-time mapping of evoked neurotransmitter release in living animals" for consideration by *eLife*. Your article has been reviewed by Ronald Calabrese as the Senior Editor, a Reviewing Editor, and two reviewers. The reviewers have opted to remain anonymous.

The reviewers have discussed the reviews with one another and the Reviewing Editor has drafted this decision to help you prepare a revised submission.

Summary:

The authors demonstrate the use of organic electrochemical transistors for detection of neurotransmitter (dopamine) release in vivo with good sensitivity and spatial/temporal resolution. The scalability of this technology and further functionalization of these transistor surfaces for different purposes can lead to several interesting applications in neuroscience.

Essential revisions:

Please explain the relative advantage of these sensors with respect to fluorescence protein based neurotransmitter sensors?

Please explain the selectivity mechanism of Pt gate for certain neurotransmitters and how this selectivity can be further improved/customized.

Please explain how you differentiate evoked electrical activity vs. neurotransmitter release (i.e. neurochemical reactions).

Please measure the reversibility of response from high to low concentration as well.

Is there a cortical damage in Figure 4—figure supplement 4 due to implantation procedure or is the missing part of cortex in Figure 4—figure supplement 4 due to tissue slice preparation?

Please highlight/summarize the novel findings regarding VTA/SNc interaction also in the introduction and mention the potential caveats of using electrical stimulation.

If feasible, please optogenetically demonstrate the existence and the effects of connectivity between VTA and SNc to support your novel findings.

*Reviewer #1:*

In this work, Shi et al., described an organic electrochemical transistor array for monitoring catecholamine neurotransmitters in rat brains. Based on the intrinsic amplification capability, the sensor allows real-time and direct readout of transient CA-NT release with a sensitivity of nanomolar range and a temporal resolution of several milliseconds. The sensor is examined in animal study for multiple site dopamine sensing, the immune response is also carefully characterized.

*Reviewer #2:*

In this manuscript, Xie et al., describe organic electrochemical transistor (OECT) arrays for measuring acute in vivo release of neurotransmitters in striatal brain regions. They fabricated a probe with 4 OECTs on a 200µm thick and 1mm wide PET film. The gate of the OECT is made out of platinum while the channel is made out of PEDOT:PSS. Authors claim that they were able to electrically stimulate different regions of the striatal pathway to evoke dopamine release and observe these fluctuations reflected in the drain current of the OECT. While this an interesting idea and potentially beneficial, there are several major concerns in the current state of the manuscript:

1) What is the mechanism of selectivity for certain neurotransmitters? It is important to provide physical view of why a Pt gate would have selectivity toward certain neurotransmitters.

2) There is no systemic electrical characterization of the transistors. Authors should provide output characteristics and modulation response without ID normalization. This will help the reader to understand the magnitude of the transistor response for defined analyte concentration.

3) The authors describe that oxidization of NTs will provide 2E- to Vg; hence the drain current decreases with increase of NT concentration (Figure 1). It is important to characterize the opposite path (high to low concentration) to be able to define the hysteresis and accuracy of measurements in Figure 2C.

4) The authors used ASCF to evaluate the detection function of their device in the presence of fixed ascorbate acid concentration. While such measurements demonstrate the detection capacity of NTs in the ACSF, it does not demonstrate whether the detection is selective to NTs. It would be necessary to have a similar measurement to Figure 2C for ascorbate acid to be able to draw conclusions regarding the selectivity of their sensor.

5) The authors provided interesting electrophysiological recordings in response to stimulation of various regions to verify the efficacy of the stimulation. While the OECT can act as a biosensor (via redox processes), the drain signal can also be modulated by electrical activity such as these evoked responses. The authors must be able to differentiate these two processes in their experimental set-up.

6) What is the spatial scale of dopamine release? There are 4 transistors on the shank -is there any spatial or temporal pattern differences between their responses that can be used for validated these recordings?

7) The histological results presented in Figure 4—figure supplement 4 show major tissue damage due to the implantation procedure. Almost the entire cortex is gone and the tract toward the VTA is damaged. Such a level of damage significantly reduces the applicability and reliability of their approach for systemic in vivo experiments.

---

## [Author Response]

Summary:The authors demonstrate the use of organic electrochemical transistors for detection of neurotransmitter (dopamine) release in vivo with good sensitivity and spatial/temporal resolution. The scalability of this technology and further functionalization of these transistor surfaces for different purposes can lead to several interesting applications in neuroscience.

We thank the editors for facilitating the review process and their efforts to compile this comment summary.

Essential revisions:Please explain the relative advantage of these sensors with respect to fluorescence protein based neurotransmitter sensors?

Recently, genetic approach has been reported to measure dopamine release in vivo. The sequence containing an encoding part for a fluorescent reporter (e.g., GFP) and an encoding part for dopamine receptors (e.g., D1 receptor, D2 receptor) is expressed in an animal brain, so that fluorescent signal would change upon dopamine binding to the receptors. To detect such fluorescent fluctuations in vivo, multi-photon confocal microscopy or fiber photometry is usually used (Tommaso et al., 2018; Sun et al., 2018; Sych et al., 2019). In comparison, the major advantages of our OECT-based technique are:

1) Quantification capability

Similar to calcium imaging based on fluorescence measurement, the protein sensors indicate dopamine fluctuations by a fractional change of the fluorescence signals (ΔF/F), which is more or less qualitative, instead of quantitative, due to the lack of a calibration of the baseline level. In the OECT-based measurement, each sensing unit can be calibrated to convert the amplitude of recorded electrical signals to absolute reading of the change of dopamine concentrations by using a pre-acquired standard curve. Such quantification capability is essential to make comparison and statistical analysis across different experiments and among different animals, and also makes the results more comparable to those in the large number of literature reports using cyclic voltammetry, or other quantitative detection methods.

2) Convenience and ease-of-use

For florescence protein-based sensors, it requires the expression of relevant proteins in the experimental animals, usually by virus mediated transfection procedures, which involve extra surgical operations and prolonged experimental preparation (3 ~ 4 weeks) for stable gene expression prior to neurotransmitter sensing. The surgical operations and gene expression may also introduce additional experimental uncertainty. In contrast, the calibrated OECT-array can be directly used with a single implantation procedure, which is more convenient and straightforward to use. The OECT technique is fully compatible with micro-fabrication techniques, which allow largescale mass production and also enable a flexible design of device configuration, such as the demonstrated multi-electrode array format for large-area spatial neurotransmitter mapping. Moreover, OECT does not require any sophisticated and expensive optical instruments that are typically needed for fluorescence detection. These features make our OECT-based method a substantially more convenient solution for the detection of neurotransmitters in vivo.

3) Potential for interference-free behavioural experiment

For florescence protein-based sensors, the detection of fluorescence fluctuation in relevant brain region is typically recorded by multi-photon microscopy or by fibre photometry systems. From an operational perspective, it is difficult to carry out multi-photon microscopy in freely moving animals. Fibre photometry devices provide a relatively more flexible alternative, but they typically require tethered cables due to a need for the delivery of excitation light and the collection of emission fluorescence from the examined brain regions, which would cause significant interference in behavioural experiments, especially in experiments studying animal social activity (Siciliano et al., 2019). For OECT, all the signals are electrical in nature, and can be easily implemented as a wireless system with the inclusion of a remote transmission module, bearing great potential for interference-free behavioural experiments.

We have now included relevant discussion in the revised manuscript (Introduction; Discussion section).

Please explain the selectivity mechanism of Pt gate for certain neurotransmitters and how this selectivity can be further improved/customized.

At the current stage, the OECT-device is employed to measure one group of neurotransmitters, the catecholamine family, including dopamine, adrenaline, and noradrenaline. These neurotransmitters all have a functional catechol group, which has a low redox potential value and high redox activity compared to other major neurotransmitters (e.g., GABA, glutamate). Therefore, the redox reaction of catecholamine NTs happens much easier on Pt electrode than that from other electroactive species in the surrounding environment(Tang et al., 2011). Our ex vivo experiments with different neurotransmitters, including GABA, glutamate, and three different catecholamines, confirmed this reasoning and showed that the OECT-array with a Pt gate gives significantly stronger response to the release of catecholamines than others (Figure 2—figure supplement 2). In this proof-of-concept animal study, we also choose to demonstrate the OECT sensing capability within the dopaminergic signaling system, in which dopamine is the major catecholamine released upon neural stimulation.

With further development, the selectivity of the OECT device could be improved by different strategies. For example, surface of the Pt gate electrode can be modified with a layer of biocompatible polymer, Nafion(Liao et al., 2014), which is a polyelectrolyte with stable Telfon backbone and acidic sulfonic groups, and is negatively charged in neutral solutions. This modification could electrostatically block the anionic interferences (e.g., ascorbic acid) on the gate electrode. Also, the selectivity could be customized through a multilayer (Nafion/PANI/enzyme/graphene) modification of the Pt gate electrode(Liao et al., 2015). A particular neurotransmitter could be selectively detected by choosing the coating enzyme within the multilayer sandwich. Specifically, for dopamine, the choice could be dopamine βhydroxylase (DBH) for more selective detection. While some of these strategies have been demonstrated *in vitro* in our previous reports (Liao et al., 2015; Tang et al., 2011), and can be further optimized for in vivo applications. We have now included relevant discussion in the revised manuscript (Discussion section).

Please explain how you differentiate evoked electrical activity vs. neurotransmitter release (i.e. neurochemical reactions).

The differentiation between evoked electrical activity and neurotransmitter release is determined by the configuration of the OECT device. The performance and nature of OECT are dominated by the ratio of the capacitance of the gate/electrolyte (C_G-E_) and that of the PEDOT:PSS channel layer (C_volumetric_), which is reversely proportional to the ratio of potential drop at the two interfaces on an OECT device (Figure 1C). As reported by George Malliaras’s group, OECT could be exploited as an electrochemical sensor or an ion-to-electron converter to amplify electrical signals, with distinct design strategies for different applications (Yaghmazadeh et al., 2011; Cicoira et al., 2010).

For device used to record electrical activity, the gate/electrolyte capacitance is larger than the capacitance of channel, leading to a larger potential at the channel layer when the total voltage applied across gate/electrolyte/channel is maintained constant. Thus, change of ion concentration upon neuronal spiking activity influences the amount of cations injected into the channel (driven by the gate voltage) and affects the conductivity through doping and de-doping the channel, which favors the detection of transient electrical signals. Non-polarizable gate or gate with high capacitance (e.g., coated with organic conductive materials) was adopted for in vivo electrophysiology applicationsto guarantee that the potential drop on the channel layer is large enough to modulate channel current for amplifying electrical signals (Khodagholy et al., 2013).

For electrochemical sensing, our OECT device is more complicated because of a trade-off between chemical detection and effective gating of the transistor. In principle, chemical detection on the Pt gate demands a smaller gate/electrolyte capacitance to enhance detection sensitivity (compared to the channel/electrolyte capacitance), whereas effective gating requires vice versa. The detection of catecholamine NTs depends on the chemical oxidation on the Pt gate electrode, which is maximized according to anatomical structure of brain nucleus. When faradic current is generated at the gate/electrolyte interface as a result of electrochemical reactions, the gate/electrolyte potential decreases, the channel/electrolyte potential increases, and the channel is de-doped. Subsequently, the channel conductivity is lowered and the current flowing through channel is decreased. Theoretically, a smaller gate/electrolyte capacitance (larger gate/electrolyte potential) should be designed to provide a wider operation window for OECTs to amplify the electrochemical reaction activity. However, the gate electrode could not be miniaturized too much, in order to maintain certain potential drop on the channel and to avoid a weak gate modulation of the device.

It is therefore important to tune the capacitance ratio more carefully in this case. The capacitance of Pt gate electrode for detection of catecholamine is an electrical double layer (EDL), which is estimated to be around 4.8 nF (10 μF/cm^2^, 0.08 × 0.06 cm^2^), considering the EDL of Pt per unit area and the electrode size (Bonthuis et al., 2011; Islamet al., 2008; Drüschler et al., 2010). By tuning the size of the channel, the volumetric capacitance of PEDOT:PSS channel in electrolyte is estimated to be around 0.47 nF (39 F/cm^3^, 0.03 × 10 × 40 μm^3^), which is effective for gating to amplify signal (Rivnay et al., 2015), and the oxidation of catecholamine on the gate electrode is maximized while gating is kept effective, so as to configure the OECT as chemical sensor in favor of electrochemical detection.

Meanwhile, the differentiation can also be supplemented by analyzing the temporal dynamics of the signals, as evoked action potentials are usually recorded as fast spiking fluctuations on millisecond scale (Yaghmazadeh et al., 2011; Cicoira et al., 2010), and signals from the release of NTs typically last for seconds (Figure 3D).

We have now included relevant discussion in the revised manuscript (Discussion section).

Please measure the reversibility of response from high to low concentration as well.

In our study, electrical stimulation was used to induce pulsed dopamine release, which caused an initial burst increase and then a gradual decrease (from peak value, as a result of diffusion and metabolism) of dopamine concentration in associated extracellular space. So, both of the ascending (from low to high) and descending process (from high to low) of dopamine dynamics were captured by the OECT device and already presented in the manuscript, as shown in Figure 3D. Upon electrical stimulation, the I_DS_ curves showed an immediate downward fluctuation, indicating the burst dopamine increase; and then a gradual upward reverse towards the baseline, indicating the decrease of dopamine concentration from a peak value. The reversibility of I_DS_ was repeatedly demonstrated in Figure 4C for tens of detection trials.

Is there a cortical damage in Figure 4—figure supplement 4 due to implantation procedure or is the missing part of cortex in Figure 4—figure supplement 4 due to tissue slice preparation?

Implantation of the OECT device to VTA or striatum would not cause any cortical damage. That appeared in the original Figure 4—figure supplement 4 and was due to tissue slice preparation. This supplementary figure has now been updated in the revised manuscript.

Please highlight/summarize the novel findings regarding VTA/SNc interaction also in the introduction and mention the potential caveats of using electrical stimulation.

As suggested, we have now updated the introduction to highlight our findings regarding the VTA/SNc interaction (Introduction):

“Our study provides an electrochemical mapping in the striatal brain and also reveals a complex cross-talk between mesolimbic and nigrostriatal signalling, which relies on a reciprocal innervation between VTA and substantia nigra pars compacta (SNc) nuclei. We found that the connection between the two nuclei can significantly affects dopamine release in NAc and lower CPu, but not upper CPu, suggesting a heterogeneous mapping from SNc to CPu (Margolis et al., 2008; Lerner et al., 2015).”

We also mentioned the potential caveats of using electrical stimulation in this revision (Introduction):

“Electrophysiological methods typically lack cell selectivity and can only be used for stimulatory but not inhibitory manipulation, and their spatial resolution can be limited and highly depend on the size of electrodes (Tye and Deisseroth, 2012; Borchers et al., 2012).”

If feasible, please optogenetically demonstrate the existence and the effects of connectivity between VTA and SNc to support your novel findings.

We agree that optogenetic demonstration would be useful to further verify our findings about the existence and the effects of a mutual innervation between VTA and SNc in dopaminergic signaling. Optogenetic validation could also potentially address the stimulation specificity issue from electrical manipulation. However, mainly due to logistic issues, it is difficult to carry out a series of well-designed optogenetic experiments within the given revision period, which would be a good topic and require substantially more time for future exploration, with a combinatory use the OECT as an electrochemical characterization device. As pointed out by the Senior Editor at the full submission stage, this Tools and Resources article focuses mainly on a novel technique about in vivo application of OECT for mapping the release of neurotransmitters, but not the new science.

Actually, electrical stimulation is widely used in relevant investigations of dopamine release in vivo and remains as an effective method for neural stimulation (Garris et al., 1999; Heien et al., 2005; Phillips et al., 2003). However, it is true that the two nuclei, VTA and SNc, are frequently treated as a whole in many studies (Margolis et al., 2008; Menegas et al., 2015), though their cell compositions are dramatically different. The VTA/SNc mutual connections has recently been shown by other studies. For example, CLARITY with highly cell-specific tracing revealed that SNc-neurons projecting to dorsomedial and dorsolateral striatum receive a small portion of neural input from VTA (Lerner et al., 2015), which echoes our observations of a mutual innervation between VTA and SNc made by electrical stimulation and recording. However, in this manuscript, we cannot rule out the possibility that other indirect network or multi-synaptic transmission (Watabe-Uchida et al., 2012) contributed to the electrical recording results, and therefore have toned down our claims in the revised manuscript (Discussion section).

References:

Borchers S, et al. "Direct electrical stimulation of human cortex - the gold standard for

mapping brain functions?" Nature Reviews Neuroscience 13.1 (2012): 63.

Bonthuis DJ, et al. "Dielectric profile of interfacial water and its effect on double-layer

capacitance." Physical Review Letters 107.16 (2011): 166102.

Cicoira F, et al. "Influence of device geometry on sensor characteristics of planar organic

electrochemical transistors." Advanced Materials 22.9 (2010): 1012-1016.

Heien ML, et al. "Real-time measurement of dopamine fluctuations after cocaine in the brain of behaving rats." PNAS 102.29 (2005): 10023-10028.

Islam MM, et al. "Electrical double-layer structure in ionic liquids: a corroboration of the

theoretical model by experimental results." Journal of Physical Chemistry C 112.42 (2008):16568-16574.

Lerner TN, et al. "Intact-brain analyses reveal distinct information carried by SNc

dopamine subcircuits." Cell 162.3 (2015): 635-647.

Menegas W, et al. "Dopamine neurons projecting to the posterior striatum form an

anatomically distinct subclass." Elife 4 (2015): e10032.

Siciliano CA et al. "Leveraging calcium imaging to illuminate circuit dysfunction in addiction." Alcohol 74 (2019): 47-63.

Sun F, et al. "A genetically encoded fluorescent sensor enables rapid and specific detection of dopamine in flies, fish, and mice." Cell 174.2 (2018): 481-496.

Sych Y, et al. "High-density multi-fiber photometry for studying large-scale brain circuit dynamics." Nature methods 16.6 (2019): 553.